# The Collagens DPY-17 and SQT-3 Direct Anterior–Posterior Migration of the Q Neuroblasts in *C. elegans*

**DOI:** 10.3390/jdb9010007

**Published:** 2021-02-19

**Authors:** Angelica E. Lang, Erik A. Lundquist

**Affiliations:** Department of Molecular Biosciences, The University of Kansas, Lawrence, KS 66046, USA; aelang@ku.edu

**Keywords:** directed neuronal migration, collagen, UNC-40/DCC, PTP-3/LAR, extracellular matrix, *C. elegans*

## Abstract

Cell adhesion molecules and their extracellular ligands control morphogenetic events such as directed cell migration. The migration of neuroblasts and neural crest cells establishes the structure of the central and peripheral nervous systems. In *C. elegans*, the bilateral Q neuroblasts and their descendants undergo long-range migrations with left/right asymmetry. QR and its descendants on the right migrate anteriorly, and QL and its descendants on the left migrate posteriorly, despite identical patterns of cell division, cell death, and neuronal generation. The initial direction of protrusion of the Q cells relies on the left/right asymmetric functions of the transmembrane receptors UNC-40/DCC and PTP-3/LAR in the Q cells. Here, we show that Q cell left/right asymmetry of migration is independent of the GPA-16/Gα pathway which regulates other left/right asymmetries, including nervous system L/R asymmetry. No extracellular cue has been identified that guides initial Q anterior versus posterior migrations. We show that collagens DPY-17 and SQT-3 control initial Q direction of protrusion. Genetic interactions with UNC-40/DCC and PTP-3/LAR suggest that DPY-17 and SQT-3 drive posterior migration and might act with both receptors or in a parallel pathway. Analysis of mutants in other collagens and extracellular matrix components indicated that general perturbation of collagens and the extracellular matrix (ECM) did not result in directional defects, and that the effect of DPY-17 and SQT-3 on Q direction is specific. DPY-17 and SQT-3 are components of the cuticle, but a role in the basement membrane cannot be excluded. Possibly, DPY-17 and SQT-3 are part of a pattern in the cuticle and/or basement membrane that is oriented to the anterior–posterior axis of the animal and that is deciphered by the Q cells in a left–right asymmetric fashion. Alternatively, DPY-17 and SQT-3 might be involved in the production or stabilization of a guidance cue that directs Q migrations. In any case, these results describe a novel role for the DPY-17 and SQT-3 collagens in directing posterior Q neuroblast migration.

## 1. Introduction

The migration of neuroblasts and neurons after their birth establishes the structure of the central nervous system (e.g., the cerebral and cerebellar cortices) and peripheral nervous system (e.g., migration of neural crest cells to form the dorsal root ganglia). In *C. elegans*, the Q neuroblasts and their descendants undergo long-range migrations in a left–right (L/R) asymmetric fashion (reviewed in [1]). The bilateral Q neuroblasts are born in the posterior lateral region of the animal as the sisters of the V5 hypodermal seam cells in embryogenesis. In the L1 larva, QR on the right polarizes and migrates to the anterior before its first division, after which the QR descendants continue to migrate anteriorly and divide to produce three neurons—AQR, SDQR, and AVM (Figure 1A) [2,3,4,5]. QL on the left polarizes and migrates posteriorly; QL descendants migrate posteriorly and undergo an identical series of divisions as QR does to produce three neurons—PQR, SDQL, and PVM (Figure 1B)—that are analogous in form and function to those produced on the right from QR [2,3,4,5].

While similar in origin and development, QL and QR have an inherent L/R asymmetry that controls anterior–posterior (A/P) migration [4,6]. In QL, the transmembrane receptors UNC-40/DCC and PTP-3/LAR act in parallel to drive posterior migration, whereas in QR these molecules mutually inhibit one another’s roles in posterior migration, resulting in anterior QR polarization and migration [4]. The nature of the L/R asymmetric function of these molecules in QL versus QR is unknown, but it might involve the fat-like cadherin CDH-4 [7].

The Gα protein GPA-16 controls many L/R asymmetry events in *C. elegans*, including internal organ asymmetries of the gut and gonad, which is often reversed in *gpa-16* mutants [8,9,10]. Here, we find that Q cell A/P migration is not affected in *gpa-16* mutants, even in animals with reversed gut–gonad asymmetry. This suggests that GPA-16 is not involved in Q cell L/R asymmetry. A L/R asymmetry in the extracellular matrix (ECM) cuticle involving the chirality of collagenous cuticle fibers is also not affected by GPA-16 [11]. Possibly, Q cell and cuticle L/R asymmetry are controlled by a distinct mechanism not involving GPA-16 or by a mechanism that acts upstream of GPA-16.

Components of the ECM often serve as ligands for transmembrane guidance and cell adhesion receptors. While the transmembrane receptors UNC-40/DCC and PTP-3/LAR act as guidance receptors in Q cells, no extracellular factor has been identified that controls the direction of Q migration. Body wall muscle cells produce SPON-1/F-spondin and a signal dependent upon NFM-1/Merlin, both of which control the protrusive ability of the Q cells but do not affect direction [12,13]. Wnt ligands control Q descendant migrations [14,15,16] but do not apparently affect initial Q protrusion and migration.

Here, we report that the extracellular matrix cuticle collagens DPY-17 and SQT-3 [17,18] are required for initial Q cell direction of protrusion. Basement membrane (BM) collagens have been broadly implicated in nervous system development, cell migration, and cell adhesion in vivo and in vitro [19,20,21]. DPY-17 is most similar to human collagen alpha-1(XXIV) (COL24A1) and is composed of a single collagen triple helix repeat and a nematode cuticle collagen N-terminal domain. The function of COL24A1 is not known, although it is associated with osteoblast differentiation as expected for a collagen involved in bone formation [22]. A human GWAS study on Hallux Valgus, a heritable syndrome of foot malformation, identified an expression quantitative trait locus in COL24A1 [23], suggesting a possible role in morphogenesis. SQT-3 is most similar to human collagen alpha-1(XXI) (COL21A1) and contains two collagen triple helix repeats and a nematode cuticle collagen N-terminal domain. The role of COL21A1 is also not understood, but human GWAS studies on atypical psychosis and nonsyndromic cleft lip/palate implicate a role of COL21A1 [24,25]. Our results indicate that DPY-17 and SQT-3 collagens provide instructive directional migration information for Q neuroblast A/P migrations. Furthermore, we show that DPY-17 acts genetically with UNC-40/DCC and PTP-3/LAR to control Q neuroblast direction of migration in a manner consistent with having a role in both pathways, a novel role for an ECM collagen.

A survey of other extracellular matrix molecules for roles in Q migration direction was undertaken. We found that some components had no effect (e.g., the cuticle collagens DPY-10, DPY-13, and SQT-1), whereas some controlled the ability to migrate, but not direction (e.g., the laminina EPI-1 and the basement membrane collagen LET-2). These results point to a specific role of DPY-17 and SQT-3 collagens in directing Q migration, and not a general disruption of collagens in the ECM or general disruption of the ECM. Mutations in the basement membrane collagen gene *emb-9* caused Q directional defects, as did mutations in genes affecting the production of Heparan Sulfate Proteoglycans (HSPGs), suggesting that multiple ECM factors control Q migration. In sum, this work describes unique instructional roles for DPY-17 and SQT-3 collagens in directional cell migration in development.

## 2. Methods

Genetics: *C. elegans* were grown using standard techniques and all experiments were performed at 20 °C. The following mutations and transgenes were used: LGI: *gpa-16(it143)M+, unc-40(n324), mec-8(e398).* LGII: *dpy-10(e128), sqt-1(sc103 and e1350)*, *ptp-3(mu245), lqIs244[Pgcy-32::cfp], ayIs9[Pegl-17::gfp]* LGIII: *dpy-17(e164, e1295* and *e1345)*, *dpy-31(e2919), emb-9(b117, b189, g23, hc70,* and *g34), hse-5(tm472* and *ok2493)* LGIV: *dpy-13(e184), unc-129(ev554), epi-1(rh92, rh233,* and *rh27), col-99(ok1204)* LGV: *sqt-3(e2924, e2926*, and *sc63), dbl-1(nk3), lqIs58[Pgcy-32::cfp].* LGX: *him-4(e1266, rh319,* and *e1267), let-2(g25, g37,* and *b246), lqIs97[Pscm::mCherry].* The temperature-sensitive lethal *emb-9* and *let-2* alleles were grown at permissive 15 °C and shifted to nonpermissive 25 °C as late embryos. *gpa-16(it143)* mutants were grown at 25 °C.

L1 synchronization and Q cell imaging: L1 larvae were synchronized by hatching as previously described [5,15]. Gravid adults and larvae were washed from plates, on which many eggs had been laid. After one hour, newly hatched larvae were washed from plates and allowed to develop for 4 h (for the 4–4.5 h timepoint). L1 larvae were mounted for microscopy on 2% agarose pads containing 5mM sodium azide as a paralytic. Q cell position and morphology were imaged using the *ayIs9[Pegl-17::gfp]* or *lqIs97[Pegl-17::mCherry]* transgene and standard epifluorescence microscopy.

Scoring of AQR and PQR migration: AQR and PQR, neuronal descendants of QR and QL, respectively, migrate the longest distances of Q descendants into the head and tail of the animal. AQR migrates anteriorly to a region near the anterior deirid and the posterior pharyngeal bulb. PQR migrates posteriorly past the anus into the phasmid ganglia in the tail. AQR and PQR positions in young adult animals were scored as previously described [5,15]. The animal was divided into five regions along the anterior–posterior axis: position 1 is the normal position of AQR; position 2 is posterior to position 1 but anterior to the vulva; position 3 is near the vulva; position 4 is the position of Q cell birth; position 5 is the normal position of PQR posterior to the anus. AQR and PQR positions were scored in 100 animals of each genotype (Table 1).

## 3. Results

**QL and QR left–right asymmetry of migration does not involve GPA-16.** The Q cells are born in the embryo as the sisters of the V5 epidermal seam cells. Before migration, they reside between the V4 and V5 seam cells in the posterior of the animal (Figure 1). Migrations of QL and QR display left–right (L/R) asymmetry. In wild-type, QL migrates posteriorly and divides over the epidermal seam cell V5, and QR migrates anteriorly and divides over the epidermal seam cell V4 (Figure 1A–C) [1,4,5]. Embryonic left/right (L/R) chirality is established at the four-cell stage in the ABa and ABp blastomeres [26]. This involves asymmetric L/R placement of the mitotic spindle poles in these cells by the Gα protein GPA-16 [8,9,10]. In *gpa-16(it143)* temperature-sensitive mutants, L/R asymmetric organ placement is frequently reversed, including L/R gut–gonad asymmetric placement.

We asked if QL and QR L/R asymmetries of posterior versus anterior migrations involved GPA-16. In *gpa-16(it143)* mutants at 25 °C, initial QL and QR protrusions and migrations were unaffected. QL protruded and migrated posteriorly, and QR protruded and migrated anteriorly (n = 50 for each). Furthermore, the Q descendants, AQR and PQR, migrated normally, even in animals with reversed gut–gonad L/R asymmetry (Table 1). Thus, the L/R asymmetries of QL and QR migrations does not involve GPA-16.

**The DPY-17 and SQT-3 collagens affect Q cell anterior–posterior migration.***dpy-17* encodes a cuticular collagen similar to collagen alpha-1 (XXIV) (COL24A1) [17,18]. *dpy-17(e164)*, a predicted null mutant, displayed defects in AQR and PQR migration. AQR sometimes migrated posteriorly, and PQR sometimes migrated anteriorly (Figure 2 and Table 1). The *dpy-17(e1295)* mutant displayed similar defects, and the hypomorphic *dpy-17(e1345)* mutant showed fewer defects (Table 1).

Initial Q migration was also affected by *dpy-17*. In *dpy-17(e164)*, 6/100 QL migrated anteriorly and divided over V4 (Figure 1D, and 6/100 QR migrated posteriorly and divided over V5 (Figure 1E). These data indicate that the collagen DPY-17 controls initial Q cell A/P directional protrusions and migrations.

*sqt-3* encodes a cuticular collagen similar to collagen alpha-1(XXI) (COL21A1). Genetic and biochemical studies suggest that DPY-17 and SQT-3 interact in the same collagen structures, possibly in hetero-oligomeric trimers [18]. *sqt-3* mutants *e2924* and *e2906* showed AQR and PQR reversals to a level similar to *dpy-17(e164)* (Table 1). *sqt-3(e2924)* is likely a null allele [18]. The dominant *sqt-3(sc63)* mutation results in a dominant left-handed Roller phenotype but showed no AQR or PQR migration reversals alone (Table 1).

*dpy-17(e164); sqt-3(e2924)* double mutants displayed AQR and PQR migration defects that were not significantly different than either single mutant alone (Table 1), suggesting that they act in the same pathway. This is consistent with DPY-17 and SQT-3 affecting the same collagen structures as previously reported [18]. *dpy-17(e164)* double mutants with dominant *sqt-3(sc63)* displayed significantly increased PQR migration defects compared to *dpy-17(e164)* alone (31%; Table 1), suggesting that *sqt-3(sc63)* might have a dominant interfering effect on PQR migration that is revealed in the absence of DPY-17.

Mutations in the cuticle collagen genes *dpy-10, sqt-1,* and *dpy-13* showed no AQR or PQR migration defects (Table 1), suggesting this effect on migration direction is specific to DPY-17 and SQT-3, and not a general disruption of collagen in the ECM. The DPY-31 BMP-1/Tolloid-like metalloprotease acts with SQT-3 and DPY-17 in cuticle formation and in connective tissue integrity involving the TGFβ molecule DBL-1 [17,27]. *dpy-31(e2919)* and *dbl-1(nk3)* predicted null mutants displayed no AQR or PQR migration defects, nor did *unc-129(ev554)*, which is predicted to encode a TGFβ-like molecule that controls axon guidance [28] (Table 1). These data suggest that the role of DPY-17 in directed Q migration is independent of DPY-31, DBL-1, and UNC-129, although redundancy of function among these molecules cannot be excluded.

**DPY-17 and SQT-3 interact genetically with UNC-40 and PTP-3**. The transmembrane receptors UNC-40/DCC and PTP-3/LAR control Q directional migration. *unc-40/DCC* and *ptp-3/LAR* mutants show defects in QL and QR migrations such that QL sometimes polarizes and migrates anteriorly, and QR sometimes polarizes and migrates posteriorly [4,6,7]. This results in misdirected Q descendant AQR and PQR migration (Table 1). In *unc-40; ptp-3* double mutants, posterior migration was nearly completely abolished, with most QL/QR and AQR/PQR cells migrating to the anterior [4]. This led to the idea of inherent L/R asymmetries of UNC-40/DCC and PTP-3/LAR functions; in QL, UNC-40/DCC and PTP-3/LAR act in parallel to drive posterior migration, and in QR, they mutually inhibit each other to prevent posterior migration [4].

In *dpy-17(e164); unc-40(n324)* double mutants, defective anterior migration of PQR was significantly enhanced (Table 1: 44% wild-type position 5 compared to 16%; 48% in position 1 in the anterior compared to 80%). *dpy-17(e1295); unc-40(n324)* showed a similar significant effect, but hypomorphic *dpy-17(e1345); unc-40* did not (Table 1). *sqt-3(e2924); unc-40(n324)* double mutants also showed significant enhancement of PQR migration defects. These results suggest that DPY-17 and SQT-3 act in parallel to UNC-40/DCC in posterior migration.

*dpy-17(e164); ptp-3(mu245)* doubles showed significantly enhanced PQR anterior migration defects, but to a lesser extent than *dpy-17; unc-40*. Defective AQR posterior migration was enhanced in *dpy-17; ptp-3* (*p* = 0.052), an effect not observed in *dpy-17; unc-40*. These results indicate that DPY-17 might act in parallel to UNC-40 in posterior PQR migration. Interaction with PTP-3 was more complex, with DPY-17 potentially acting in parallel for both posterior PQR migration and anterior AQR migration.

**EPI-1/lamininα5 and HIM-4/hemicentin affect the ability of AQR and PQR to migrate.** EPI-1 and HIM-4 are extracellular matrix constituents known to control cell migration [29,30,31,32,33,34,35]. *epi-1* mutants displayed high-frequency defects in AQR and PQR migration, but few directional defects (Table 1). In other words, *epi-1* affected the ability of AQR and PQR to execute their migration but did not affect direction of migration. *epi-1(rh233)* did not enhance directional migration defects of *unc-40(n324)* (Table 1). Rather, the double mutant displayed an additive phenotype in which misdirected cells failed to complete their migrations at a higher frequency (i.e., in *unc-40(n324)*, 48% of misdirected PQRs migrated completely anteriorly to position 1, whereas only 13% did so in *epi-1(rh233); unc-40(n324)* double mutants, *p* < 0.0001). *him-4* mutants displayed low-frequency failures in AQR and PQR migration. One allele of three analyzed displayed one misdirected PQR in position 3 in the midbody. One allele, *him-4(rh319)*, significantly enhanced PQR defects of *unc-40* and *ptp-3*, but another, *him-4(e1266),* did not significantly affect AQR and PQR migration defects of *unc-40* or *ptp-3*. Together, these data suggest that EPI-1 is required for the ability of cells to migrate but is not involved in the direction of migration. HIM-4 also controls the ability of cells to migrate but might also have a role in PQR direction of migration, although the effect is not as strong as DPY-17.

**EMB-9/collagenIVα5 controls PQR directional migration.***emb-9* mutants display a variety of developmental defects, including defective migration of the gonadal distal tip cells [36,37,38]. Three temperature-sensitive *emb-9* mutants displayed low-frequency reversals of PQR migration, as well as incomplete AQR migration (Table 1). LET-2/CollagenIVa5/6 also regulates gonadal distal tip cell migration [37,39,40]. *let-2* mutants showed low-frequency failures of AQR and PQR migration, but no directional defects (Table 1). Previous studies showed the basement membrane collagen CLE-1/collagenXVIII, which organizes nervous system and synapse structure [41,42] had no role in Q migration [43].

*C. elegans* COL-99/COL13A1 is a member of the conserved Membrane-Associated Collagens with Interrupted Triple Helices (MACIT) family of collagens [44,45], which contain a transmembrane domain. COL-99 regulates axon fasciculation, outgrowth, and ventral nerve cord left–right asymmetry [45,46]. *col-99* mutants displayed no defects in AQR or PQR migration (Table 1).

**HSE-5 acts in parallel to UNC-40/DCC in posterior PQR migration**. The heparan sulfate epimerase HSE-5 was previously shown to control directional QR and QL migration and AQR/PQR migration [43,47] (Table 1). *hse-5* mutants significantly enhanced PQR directional migration defects of *unc-40*, and significantly enhanced AQR directional defects of *ptp-3*. *hse-5* significantly suppressed PQR directional defects of *ptp-3*. This complex genetic interaction could reflect the role of HSE-5 in modifying the function of multiple extracellular heparan sulfate proteoglycans with distinct roles in Q migration.

**MEC-8/RBPMS controls directional AQR/PQR migration.** The MEC-8 RNA binding protein and mRNA processing factor is involved in the processing of mRNAs, including that of the extracellular heparan sulfate proteoglycan UNC-52/perlecan, which is involved in gonadal distal tip cell migration [48,49,50,51]. *mec-8(e398)* displayed low-frequency defects in AQR and PQR directional migration but did not significantly enhance *unc-40* or *ptp-3* (Table 1). In fact, *mec-8* significantly suppressed PQR defects of *ptp-3*, similar to *hse-5*.

## 4. Discussion

**Q cell L/R asymmetry is independent of GPA-16.** Here, we have found that the L/R asymmetry of migration of the Q neuroblasts is independent of a known mechanism or embryonic L/R asymmetry involving the Gα protein GPA-16 [8,9,10]. QR descends from the right embryonic blastomere ABpr, and QL from the left ABpl, the very L/R asymmetry that GPA-16 controls in the early embryo. However, GPA-16 does not affect Q cell L/R asymmetric migration. Previous studies showed that the extracellular matrix cuticle displays L/R chirality which is also not dependent on GPA-16 [11]. The *C. elegans* cuticle is composed of collagens and other extracellular matrix molecules [52]. The cuticle displays L/R chirality, as seen in the chiral orientation of cuticle fibers in electron micrographs and the L/R chirality of *Roller* mutants, which affect cuticular components including collagens [11]. Cuticular L/R chirality, including that of *Roller* mutants, is not affected by *gpa-16* mutation, suggesting a distinct mechanism not involving GPA-16 [11]. Possibly, whatever mechanism controls cuticular L/R chirality might also control Q neuroblast L/R asymmetry of migration. This mechanism could act in parallel to GPA-16 to control the L/R asymmetry of distinct tissues, or the ECM chirality mechanism might be earlier than the GPA-16 mechanism, such that GPA-16 is a ramification of the earlier event that controls both ECM chirality and GPA-16-mediated chirality.

**The cuticular collagens DPY-17 and SQT-3 control direction of Q migration.** We found that mutations in two genes encoding cuticular collagens, *dpy-17* and *sqt-3*, control anterior–posterior direction of migration of the Q neuroblasts and of their descendants, AQR and PQR. That the *dpy-17; sqt-3* double null mutant resembles each single mutant alone suggests that DPY-17 and SQT-3 act together in the same pathway, consistent with previous results suggesting that they might contribute to the same collagen structures, possibly in heterotrimers [18].

While Wnts control anterior–posterior migration of the Q descendants [14,15,53,54], they do not control the initial direction of Q migration. SPON-1/F-spondin and a signal produced in muscles by NFM-1/Merlin that interacts with SLT-1/Slit control the ability of Q cells to migrate, but are not involved in direction of migration [12,13]. The transmembrane fat-like cadherin CDH-4 acts non-autonomously to control Q cell direction [7], but DPY-17 and SQT-3 collagens represent the first extracellular molecules identified that control direction of initial Q migration.

A third collagen, EMB-9, also controlled AQR and PQR directions of migration, but the collagen LET-2 did not. Furthermore, mutations in three additional cuticular collagen genes, *dpy-10, dpy-13,* and *sqt-1,* did not affect AQR/PQR migration, indicating that general collagen perturbation was not the cause of Q migration defects in *dpy-17* and *sqt-3*, but rather a specific role of DPY-17 and SQT-3 in controlling direction of initial Q migration and AQR/PQR migration.

**How might collagens control Q cell direction of migration?** DPY-17 and SQT-3 are cuticular collagens that each contain the nematode cuticular collagen domain. How might the cuticle guide anterior–posterior migration? Possibly, L/R chirality in the cuticle itself [11] might provide guidance information to the Q cells which, after birth, are in contact with the cuticle. Alternatively, DPY-17 and SQT-3 might also be components of the extracellular matrix basement membrane that separates the Q cells and other hypodermis from muscle. The basement membrane serves as a substrate for cell and growth cone migration, including the Q cells, which migrate beneath the hypodermis. In support of this idea, DPY-17 acts with the fibrillin MUA-3 to maintain connective tissue function and organellar adhesion involving the basement membrane [27]. Possibly, DPY-17 and SQT-3 are part of an A/P pattern in the basement membrane tied to the anterior–posterior embryonic axis (Figure 3). One intriguing idea is that collagen fibers in the basement membrane align with the A/P axis of the animal, and that QL and QR might respond differently to this potential A/P pattern in the basement membrane. A third possibility is that DPY-17 and SQT-3 might be involved in the production of an unidentified extracellular guidance cue that controls direction. Indeed, DPY-17 acts with the DPY-31/Tolloid metalloproteinase to generate a DBl-1/TGFβ growth factor in a model of Marfan disease [27,55]. However, neither DPY-31 nor DBL-1 affected Q directional migration. Another potential mechanism involves a cuticular pattern established by DPY-17 and SQT-3 that is transmitted to the basement membrane. This idea is exemplified by *Roller* mutations that introduce left- or right-handed helical twists in the body of the animal [52]. *Roller* mutations are often in genes that encode cuticle collagens and result in a twist in the cuticle due to defects in collagen processing and crosslinking [52,56]. Indeed, *sqt-3(sc63)* is a dominant right-hand Roller [17]. That internal organs such as muscles and gut, which have no direct contact with the cuticle, are also twisted in Roller mutants indicates that the cuticular twist is reflected in the basement membrane, upon which internal organs are situated. Possibly, a pattern in the cuticle involving DPY-17 and SQT-3 is transmitted to the basement membrane, which influences direction of Q migration.

**DPY-17 and SQT-3 interact genetically with the UNC-40/DCC and PTP-3/LAR receptors.** UNC-40/DCC and PTP-3/LAR are transmembrane receptors that act in the Q cells and control initial direction of migration [4,6,57]. In QL, UNC-40 and PTP-3 act redundantly to drive posterior migration; in QR, they mutually inhibit one another, allowing for anterior migration [4]. The fat-like cadherin CDH-4 might regulate this response [7]. *dpy-17* and *sqt-3* enhance PQR directional migration defects of *unc-40* in a manner similar to *ptp-3*, suggesting that DPY-17/SQT-3 might act in the PTP-3 pathway. However, *dpy-17* also enhanced PQR defects of *ptp-3,* suggesting it functions at least partially in parallel to PTP-3 as well. Possibly, DPY-17 acts in both pathways or defines a third pathway in posterior migration. *dpy-17* also enhanced AQR defects of *ptp-3*, a phenotype not observed in *dpy-17; unc-40* doubles. DPY-17 might act in parallel to PTP-3 in anterior AQR migration, potentially in mutual inhibition of UNC-40/DCC, resulting in increased posterior migration of AQR. The L/R asymmetric function of UNC-40/DCC and PTP-3/LAR might control how the Q cells respond to the DPY-17/SQT-3 pattern (Figure 3).

**EPI-1/laminin and HIM-4/hemicentin affect migration, but not direction.***epi-1* and *him-4* mutants displayed defects in the ability of AQR and PQR to complete their migrations but did not affect direction as *dpy-17* and *sqt-3* did. The additive phenotype of *epi-1; unc-40* double mutants is consistent with a role in migration but not direction. These results suggest that the extracellular matrix serves as a source of guidance information for migrating cells (DPY-17, SQT-3), as well as permissive substrate required for migration (EPI-1, HIM-4). This reinforces the notion that DPY-17 and SQT-3 have specific roles in directing cell migration, and that general perturbations to the ECM do not cause directional defects.

**Factors regulating heparan sulfate proteoglycans control Q cell direction of migration.** The heparan sulfate epimerase HSE-5 is predicted to catalyze the epimerization of d-glucuronic acid (GlcA) to l-iduronic acid (IdoA) in the heparan side chains of heparan sulfate proteoglycans. HSE-5 was previously shown to control directional QR and QL migration and AQR/PQR migration [43,47]. *hse-5* displayed complex genetic interactions with *unc-40* and *ptp-3*, consistent with the possibility of HSE-5 acting with multiple HSPGs in different aspects of AQR and PQR migration.

The RNA processing factor MEC-8 regulates the alternative splicing of *unc-52*, which encodes the basement membrane HSPG UNC-52/perlecan, with functional consequences on body wall muscle development [51]. *mec-8* mutants displayed low-frequency defects in AQR/PQR migration direction but did not enhance *unc-40* or *ptp-3* significantly. However, *mec-8* suppressed PQR migration defects of *ptp-3* similar to *hse-5*. Possibly, UNC-52 controls AQR/PQR directional migration. Hypomorphic *unc-52* alleles, caused by mutations in exons affected by *mec-8*, did not cause AQR or PQR defects [43], and *unc-52* null mutants arrested before *gcy-32::cfp* expression in AQR and PQR (data not shown). No single or double mutant in genes encoding HSPGs resulted in AQR/PQR directional defects [43], but it remains possible that UNC-52/perlecan is involved.

The results presented here show that the collagens DPY-17 and SQT-3 instruct A/P migration of the Q neuroblasts. An intriguing idea is that DPY-17 and SQT-3 might be part of a pattern in the A/P axis of the ECM that instructs Q migration, either in the basement membrane or in the cuticle. Alternately, they could be involved in the production of an as-yet unidentified guidance cue that instructs Q migrations. In any event, these results demonstrate that collagens are involved in providing instructional guidance information to neuroblasts in vivo.

## Figures and Tables

**Figure 1 jdb-09-00007-f001:**
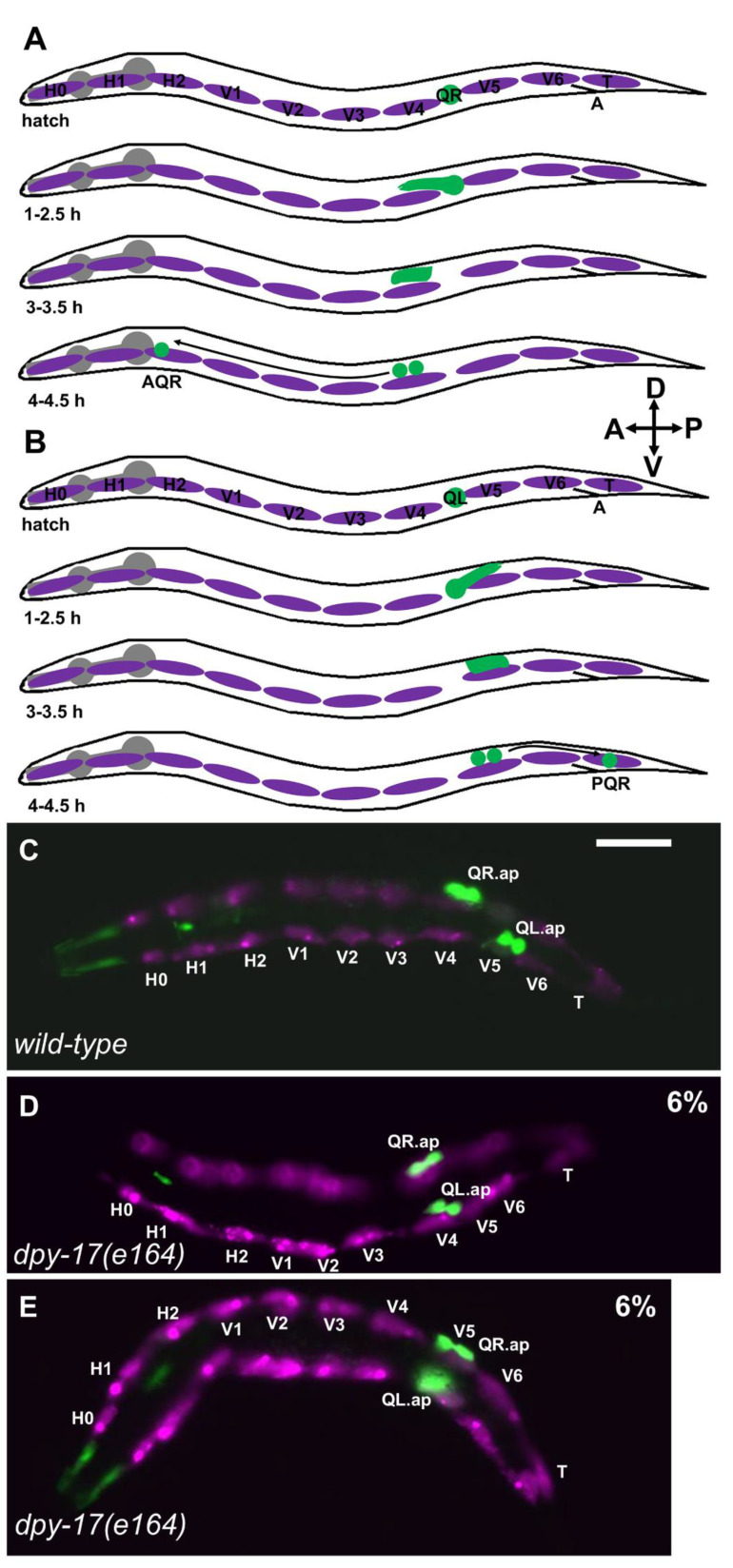
**Q neuroblast migration**. (**A**) A schematic representation of QR and descendant migration (after [4,5]). Anterior left dorsal is up, as shown by the compass in the figure. Q cells and descendants are green; the hypodermal seam cells are magenta; the pharynx is grey. “A” is the position of the anus. Timepoints after hatching are indicated. QR is born in embryogenesis as the sister of the V5 seam cell and resides between the V4 and V5 seam cells at hatching. At 1–2.5 h, QR protrudes anteriorly over the V4 seam cell. At 3–3.5 h, the QR cell body migrates anteriorly along the protrusion to reside above V4. At 4–4.5 h, QR undergoes the first cell division, to produce AQR, which migrates anteriorly to the anterior deirid ganglion near the pharynx. (**B**) Schematic of QL migration as described in (A). At 1–2.5 h, QL protrudes posteriorly over V5. At 3–3.5 h, the QL cell body migrates posteriorly along the protrusion to reside over V5. At 4–4.5 h, QL undergoes the first division, and subsequent divisions and posterior migrations result in PQR residing in the tail in the phasmid ganglion. (**C–E**) Fluorescence micrographs of L1 larvae 4–4.5 h posthatching. The Q neuroblasts are marked with *ayIs9[Pegl-17::gfp]* (green; also in unidentified cells around the pharynx). The seam cells are marked with *lqIs97[Pscm::mCherry]* (magenta). The seam cells and Q cells are indicated. The scale bar in (**C**) represents 5 μm. (**C**) A wild-type animal. QR migrated anteriorly and divided over V4; QL migrated posteriorly and divided over V5. (**D**) In a *dpy-17(e164)* mutant, QL migrated anteriorly and divided over V4 instead of V5. (**E**) In a *dpy-17(e164)* mutant, QR migrated posteriorly and divided over V5 instead of V4. QR and QL migration was defective in 6% of *dpy-17(e164)* animals (n = 100).

**Figure 2 jdb-09-00007-f002:**
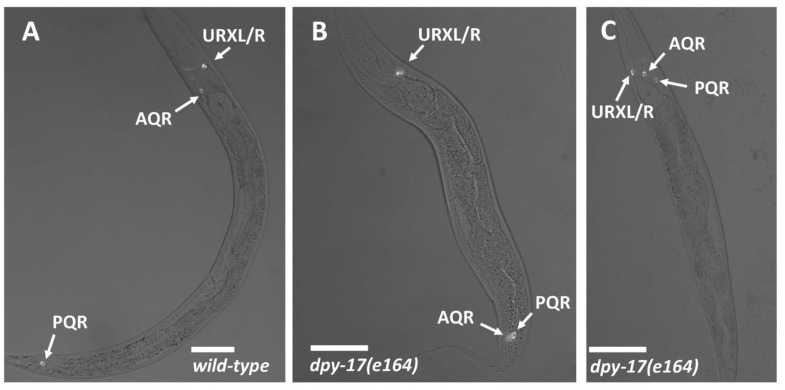
**AQR and PQR positions.** Micrographs of L4 animals expressing *lqIs244[Pgcy-32::cfp]* in AQR, PQR, and the left and right URX neurons, which reside next to one another and are indistinguishable in these micrographs. Scale bars represent 20μm. Fluorescence images are merged with differential contrast interference (DIC) images. (**A**) In wild-type, AQR resides in the anterior in the anterior deirid near the pharynx, and PQR resides posterior to the anus in the phasmid ganglion. (**B**) In a *dpy-17(e164)* mutant, AQR had migrated posteriorly near to the normal position of PQR. (**C**) In a *dpy-17(e164)* mutant, PQR migrated anteriorly to reside near AQR.

**Figure 3 jdb-09-00007-f003:**
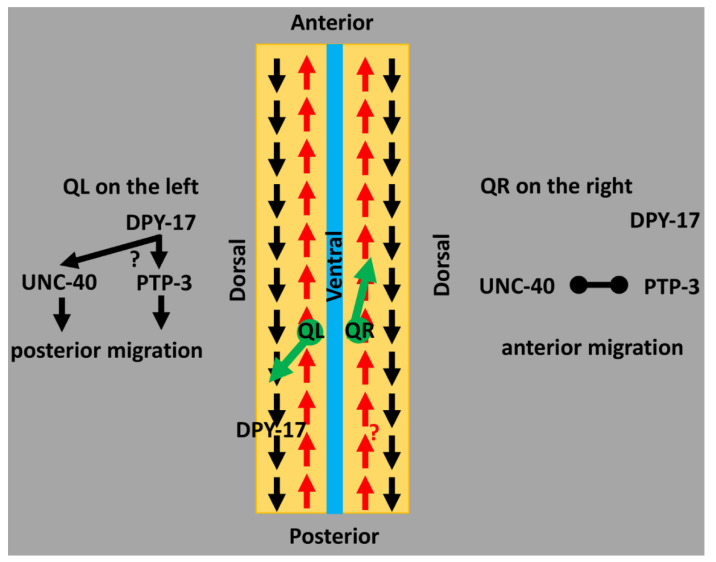
**A model of DPY-17 and SQT-3 collagens in directed neuroblast migration.** A schematic representation of an animal that has been cut along the dorsal axis and splayed open. Arrows represent a potential asymmetry in the extracellular matrix involving collagens (i.e., a pattern in the extracellular matrix (ECM) oriented to the anterior–posterior axis of the animal). The black “posterior” pattern might be defined by DPY-17 and SQT-3. In QL, this pattern is used for posterior migration. An unidentified red “anterior” pattern might be followed by QR for anterior migration. The inherent L/R asymmetry in the Q neuroblasts, possibly involving UNC-40/DCC and PTP-3/LAR receptor functions, might determine the response of the Q neuroblasts to these ECM patterns. In QL, DPY-17 might instruct UNC-40 and PTP-3 to migrate to the posterior. In QR, UNC-40 and PTP-3 inhibit one another, allowing for the “anterior” pattern to be followed.

**Table 1 jdb-09-00007-t001:**
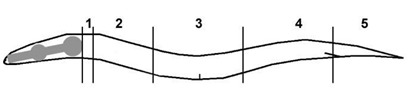
AQR and PQR migration defects.

	AQR Position (%)					PQR Position (%)				
Genotype (n = 100)	1	2	3	4	5	1	2	3	4	5
*wild-type*	100	0	0	0	0	0	0	0	0	100
*gpa-16(it143) 25 °C*	99	1	0	0	0	0	0	0	1	99
*gpa-16(it143) 25 °C (reversed)*	100	0	0	0	0	0	0	0	0	100
*dpy-17(e164)*	98	1	0	0	1	9	2	0	2	87
*dpy-17(e1295)*	98	1	0	0	1	9	0	4	0	87
*dpy-17(e1345)*	99	1	0	0	0	1	0	0	0	99
*sqt-3(e2924)*	94	3	0	1	2	5	0	1	10	84
*sqt-3(e2906)*	87	10	0	0	3	8	3	0	0	90
*sqt-3(sc63)*	98	2	0	0	0	0	0	0	0	100
*dpy-17(e164); sqt-3(e2924)*	91	6	0	0	3	9	2	2	0	87
*dpy-17(e164); sqt-3(sc63)*	97	0	0	0	3	31 **	0	0	0	69
*dpy-10(e128)*	100	0	0	0	0	0	0	0	0	100
*dpy-13(e184)*	100	0	0	0	0	0	0	0	0	100
*sqt-1(sc103)*	100	0	0	0	0	0	0	0	0	100
*sqt-1(e1350)*	100	0	0	0	0	0	0	0	0	100
*dpy-31(e2919)*	100	0	0	0	0	0	0	0	0	100
*dbl-1(nk3)*	100	0	0	0	0	0	0	0	0	100
*unc-129(ev554)*	100	0	0	0	0	0	0	0	0	100
*unc-40(n324)*	99	0	0	1	0	48	2	2	4	44
*ptp-3(mu245)*	96	0	0	1	3	33	3	2	3	59
*dpy-17(e164); unc-40(n324)*	99	1	0	0	0	80 ***	0	0	4	16 ***
*dpy-17(e1295); unc-40(n324)*	99	0	0	0	1	82 ***	1	3	2	12 ***
*dpy-17(e1345); unc-40(n324)*	94	1	0	0	5	49	3	3	0	45
*sqt-3(e2924); unc-40(n324)*	99	1	0	0	0	82 ***	3	3	3	9 ***
*dpy-17(e164); ptp-3(mu245)*	77	5	2	1	15 *	57	2	4	2	35 **
*epi-1(rh92)*	6	29	42	23	0	0	1	7	80	12
*epi-1(rh233)*	81	11	6	2	0	0	0	0	34	66
*epi-1(rh27)*	13	21	44	22	0	0	1	5	81	13
*epi-1(rh233); unc-40(n324)*	76	17	5	2	0	13 ***	9	6	14	58
*him-4(e1266)*	95	4	0	0	1	0	0	0	1	99
*him-4(e1267)*	96	3	1	0	0	0	1	0	1	98
*him-4(rh319)*	99	1	0	1	0	0	0	0	1	99
*him-4(e1266); unc-40(n324)*	98	1	0	2	1	25	1	4	15	55
*him-4(rh319); unc-40(n324)*	94	4	1	0	1	49 **	6	6	6	33
*him-4(e1266); ptp-3(mu245)*	93	2	0	2	3	31	0	1	2	66
*him-4(rh319); ptp-3(mu245)*	93	1	0	2	4	54 **	1	1	1	43
*emb-9(b117)*	99	0	0	1	0	0	0	0	0	100
*emb-9(b189)*	98	1	1	0	0	0	0	0	2	98
*emb-9(g23)*	99	0	1	0	0	1	0	1	3	95
*emb-9(hc70)*	100	0	0	0	0	2	0	0	0	98
*emb-9(g34)*	100	0	0	0	0	1	0	0	2	97
*let-2(g25)*	100	0	0	0	0	0	0	0	0	100
*let-2(g37)*	95	5	0	0	0	0	0	1	0	99
*let-2(b246)*	88	12	0	0	0	0	0	0	0	100
*col-99(ok1204)*	100	0	0	0	0	0	0	0	0	100
*hse-5(tm472)*	85	7	5	0	3	18	2	0	2	78
*hse-5(ok2493)*	85	4	1	0	10	15	2	2	7	77
*hse-5(tm472); unc-40(n324)*	65	16	15	1	3	44	17	26	6	7
*hse-5(ok2493); unc-40(n324)*	71	13	6	7	3	41	18	16	13	12
*hse-5(tm472); ptp-3(mu245)*	45	1	3	8	43 ***	12	1	4	8	75 **
*hse-5(ok2493); ptp-3(mu245)*	75	4	1	1	19	9	2	0	1	88
*mec-8(e398)*	97	2	0	0	1	1	0	0	0	99
*mec-8(e398) unc-40(n324)*	95	2	0	0	3	36	8	4	3	49
*mec-8(e398); ptp-3(mu245)*	96	1	0	2	1	11	1	1	3	84 ***

* *p* = 0.052; ** *p* ≤ 0.034; *** *p* ≤ 0.0001.

## Data Availability

All data are contained in this article.

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
