# Peer review of "The Collagens DPY-17 and SQT-3 Direct Anterior–Posterior Migration of the Q Neuroblasts in C. elegans"

_jdb, 2021, doi:10.3390/jdb9010007_

Round 1

Reviewer 1 Report

Lang and Lundquist present a manuscript using classical genetics to characterise a new role for collagens in directing neuroblast migration. They provide some intriguing hypotheses as to how this may be carried out. The work is scientifically sound, well presented, and I have no major comments. Below are some minor comments that are mostly textual and/ or intended to improve the clarity of the presentation. The authors are welcome to ignore any of these suggestions at their discretion.

Methods:

  • It would be useful to have a more detailed description of the imaging conditions. The current manuscript reads ‘mounted for microscopy’, but would be better to expand it to say x% agarose pads with x chemical paralytic, and the general type of microscope used (looks like a regular epifluorescent microscope?)
  • Was the AQR/PQR scoring done at L4?

Table 1: It might be clearer to put AQR and PQR headings over the left and the right of the table, splitting the table down the middle, to make it clear left corresponds to AQR and right to PQR

Some of the genetic nomenclature doesn’t fit convention. Gene names are normally written in order, e.g. in table 1 it reads dpy-17(e164); unc-40(n324), but dpy-17 is on III and unc-40 on I, so should read unc-40(n324); dpy-17(e164). Also, mec-8 and unc-40 are on the same chromosome so they should be written without the semi-colon: unc-40(n324)mec-8(e398)

Figure 1:

  • For A/B, nice diagrams. Having a little compass showing anterior, posterior, ventral and dorsal might help orient readers
  • C-E, it would be more intuitive to have the genotype written on the panels, in addition to the legend.
  • The legend says that V indicates the position of the vulva, but these are L1 worms! They don’t have a vulva yet
  • The scale bar is labelled in the legend as 5mm – something is off by a long way there

Figure 2: again, the scale bar says 20mm, which is off by a very long way

Results: there’s a textual error on pg 10 in the EPI-1 section: “him-4(e1266) did not. did not significantly affect…”

Figure 3: it would be much clearer to understand the model if there was some annotation on the figure itself. One can figure it out by reading the legend, but easier if it is also on the figure.

Last point: it would be nice but not necessary to have some kind of summary figure showing the interactions of the genetic/signaling pathways described in the results and where they may fit onto Q cell migration (maybe a reworked Figure 3, or a separate figure).

Reviewer 2 Report

The authors investigated the regulation of Q cell left/right asymmetry of neuroblast migration in C.elegans.

It is a very descriptive study, in which many genes tested but without going into detail or mechanistic. Still, this might provide nice overview and entry into deeper studies, and a novel role for the DPY-17 and SQT-3 Collagens in directing posterior Q neuroblast migration was found. 

I have some points that require attention.

Table 1 legend is missing, which needs to be provided. Maybe it is advantageous to collect the genes that cause significant changes in one table, and present another table collecting all genotypes that did not cause changes???

Like it is now, it is harder to capture the important things/genes.

What do the authors mean when writing: 

DPY-17 and SQT-3 interact genetically with UNC-40 and PTP-3?

„It does also not become clear in the discussion“. Here, collagens and membrane receptors are „investigated“. What do the authors mean with „genetic“? Obviously, similar/same pathways are affected, but I don’t see a genetic interaction. This should be either clearly described or phrased differently (in the results and discussion section)

Error in the following sentence:

„One allele, him-4(rh319), significantly enhanced PQR defects of unc-40

and ptp-3, but another, him-4(e1266) did not. did not significantly…“
